# Axis-Wise Ridge Regression Blending of Heterogeneous 2D-3D Registration Methods for Cochlear Implant Surgery

**Ziyi Wang**                                                    ZIYI.WANG.2@VANDERBILT.EDU

**Hannah Mason**                                                        HGMASON77@GMAIL.COM

**Jack Noble**                                                    JACK.NOBLE@VANDERBILT.EDU

*Dept. of Electrical and Computer Engineering, Vanderbilt University*

## Abstract

Accurate registration between intra-operative 2D surgical microscope images and pre-operative 3D CT images is critical for image-guided cochlear implant surgery, as it enables augmented reality guided surgery. Five complementary deep learning methods have been developed for this registration task, but their architectural biases lead to uneven performance across translation and rotation axes, so no single method performs best on all DoFs. We propose an axis-wise ridge regression blending strategy that divides the five methods into two groups: direct pose regression methods (M1, M2) for in-plane translation, and affine-based methods (M3–M5) for depth and rotation. Separate ridge regression branches are trained for different output groups, with each branch taking the full 6-DoF predictions from its assigned method subset to preserve cross-axis information. Under leave-one-out cross-validation on seven patient samples, the proposed method achieves mean absolute errors of 0.5821, 0.2991, and 48.2001 mm on TX, TY, and TZ, respectively, outperforming all five baseline methods. The closed-form solution requires no GPU and remains stable with as few as six training samples, making it well suited to data-limited surgical navigation.

**Keywords:** 2D-3D registration, cochlear implant surgery, surgical navigation, ridge regression, blending, pose estimation

## 1. Introduction

Cochlear implant surgery (CI) requires precise placement of an electrode array in the cochlea under intraoperative microscope guidance (Holden et al., 2013). Overlaying critical anatomy, such as the facial and cochlear nerves, onto live surgical images could provide valuable reference for surgeons (Wang et al., 2017; Labadie and Noble, 2018). To achieve this, we align a patient-specific epitympanum mesh reconstructed from preoperative CT to the 2D surgical field of view. The epitympanum provides a reliable registration target due to its distinctive anatomy, and once aligned, known spatial relationships allow other critical structures to be overlaid onto the surgical image. This 2D-3D registration task requires estimating a 6-DoF rigid pose from a single monocular image, but is inherently ill-posed due to depth ambiguity and the limited microscope field of view (Zhang et al., 2024).

Mason (Mason, 2025) proposed five deep learning methods for this task, including a direct 6-DoF regression method (RT), a segmentation-based pose recovery method using latitude-longitude-alpha (LLA) maps, and three affine transformation prediction methods. These methods are complementary, with each performing better on different degrees of freedom. Simple ensemble strategies, such as median voting and fixed global weight averaging, do not explicitly exploit this complementarity.

To address this limitation, our contributions are:

- An axis-wise ridge regression blending strategy that groups methods by architectural type and trains independent ridge regressors per DoF group, allowing each branch to leverage cross-axis correlations within its method subset.

- A leave-one-out cross-validation (LOOCV) evaluation over all seven patient samples, showing that the proposed method outperforms all five individual baselines.

## 2. Method

### 2.1. Five Baseline Methods

We build on the five methods proposed by Mason (Mason, 2025). M1 predicts a latitude-longitude-alpha (LLA) map from the surgical image and recovers pose via PnP, while M2 directly regresses rotation and translation. M3 predicts an affine transformation from the surgical image to atlas space, M4 uses a Siamese encoder to compare the input image with a normalized-pose curvature render for affine prediction, and M5 extends M4 with random warp augmentation. For M3–M5, the predicted affine transform is applied to a normalized-pose LLA render before PnP pose recovery.

The methods are complementary: M1 and M2 behave as direct image-to-pose mappings and tend to better capture in-plane translation, whereas M3–M5 rely on affine alignment cues that are often more informative for depth and rotation. This motivates partitioning them into two groups for blending.

### 2.2. Axis-Wise Ridge Regression Blending

Given the predictions of all five methods on the training set, we train two ridge regression branches: an XY branch for in-plane translation and a ZR branch for depth and rotation. The XY branch takes the normalized 6-DoF outputs of M1 and M2, while the ZR branch takes those of M3–M5. Each branch uses the complete 6-DoF outputs of its assigned methods, rather than only the target DoFs, to preserve cross-axis information.

For each branch $b \in \{xy, zr\}$ with training matrix $X_b \in \mathbb{R}^{N \times d_b}$ and target matrix $Y_b$, the ridge weight matrix is computed in closed form:

$$W_b = (X_b^\top X_b + \lambda_b I)^{-1} X_b^\top Y_b.$$

At test time, predictions are obtained as $\hat{y}_b = x_b^\top W_b$ and then denormalized for evaluation. The regularization parameters $\lambda_{xy}$ and $\lambda_{zr}$ are selected independently by grid search and are both set to 0.001.

## 3. Experiments

### 3.1. Dataset and Evaluation Protocol

The dataset comprises 12 patient sequences, each containing intraoperative microscope images, a patient-specific 3D surface mesh of the epitympanum, and corresponding ground-truth 6-DoF poses. Five sequences are used to train the five baseline methods, while the remaining seven are used for leave-one-out cross-validation (LOOCV). In each of the seven folds, six sequences are used to train the blending weights, and the remaining sequence is

held out for testing. Performance is reported as the mean absolute error (MAE) averaged over all seven folds.

## 3.2. Results and Ablation Study

We compare the proposed method with a hybrid baseline and five alternative blending strategies listed in Table 1. The hybrid baseline combines the strongest individual baseline predictions across translation axes, using M1 for TX and TY and M5 for TZ. For rotation, none of the ensemble methods outperformed M5, so we retained M5 directly.

Table 1: LOOCV MAE results on the translation degrees of freedom.

| Method | TX (mm)↓ | TY (mm)↓ | TZ (mm)↓ |
|---|---|---|---|
| Hybrid Baseline (TX/TY from M1, TZ from M5) | 0.6385 | 0.5781 | 52.8521 |
| Global Ridge | 0.6178 | 0.3171 | 48.8728 |
| Global MLP | 0.5555 | 0.4010 | 64.4599 |
| **Axis-wise Ridge (Proposed)** | **0.5821** | **0.2991** | **48.2001** |
| Axis-wise Ridge, strict | 0.5508 | 0.3036 | 63.2648 |
| Axis-wise MLP | 0.4594 | 0.3476 | 56.9447 |
| Axis-wise MLP, strict | 0.5869 | 0.3038 | 54.7371 |

The alternatives are Global Ridge and Global multilayer perceptron (MLP), which use all 30 inputs jointly; Axis-wise Ridge (strict) and Axis-wise MLP (strict), in which each branch uses only its target DoFs; and Axis-wise MLP, which keeps the same non-strict partitioning as our method but replaces ridge regression with an MLP.

The comparison focuses on translation DoFs. The proposed axis-wise ridge regression achieves the lowest MAE on TY and TZ and improves over the hybrid baseline on all three translation DoFs. More broadly, the results suggest that axis-wise partitioning is more effective than global blending for translation, ridge regression is more robust than MLP in this small-sample setting, and the non-strict design benefits from useful cross-axis context.

## 4. Conclusion

We presented an axis-wise ridge regression blending strategy for 2D-3D registration in cochlear implant surgery. By partitioning five heterogeneous methods into two groups and fitting separate closed-form ridge regressors, the proposed method outperforms all baselines on the three translation DoFs under LOOCV across seven patient samples. The method is lightweight, interpretable, and requires no GPU, making it practical in the data-limited setting of surgical navigation. Rotation accuracy, however, does not improve consistently; future work will explore improved groupings and rotation-specific features.

## Acknowledgments

This research is supported by grant R01DC022099 from the National Institute for Deafness and Communication Disorders.

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
