# OpenReview forum: "Axis-Wise Ridge Regression Blending of Heterogeneous 2D-3D Registration Methods for Cochlear Implant Surgery"
_MIDL.io/2026/Short_Papers — MIDL 2026 - Short Papers Poster_

### Official Review · Reviewer_U1Gp · 2026-04-28
**intersting clnical problem wiht prelimanry initial work**

**Rating:** 4
**Confidence:** 4

**Review:**

The paper addresses an important problem in image-guided surgery—accurate 2D–3D registration for cochlear implant procedures—where robustness across all degrees of freedom is critical. The work
focused on overcoming some of the challanges of the authors own work.  3The use of a closed-form ridge regression solution is appealing for low-data regimes and contributes to the practicality of the approach.

In terms of originality, the contribution is modest. The method primarily applies standard regression-based ensembling with a task-specific partitioning strategy across axes. While the axis-wise grouping is a reasonable and somewhat novel design choice, the overall framework remains an incremental extension of existing ensemble and regression techniques rather than a fundamentally new methodological advance.

The experimental setup is limited by the very small dataset (seven patients in LOOCV), which raises concerns about statistical robustness and generalizability. While the method outperforms the included baselines in translation error, improvements are relatively modest and inconsistent across all degrees of freedom—particularly for rotation, which remains a critical challenge. Additionally, comparisons are restricted to variations of the same baseline family, with no benchmarking against broader state-of-the-art registration approaches (not the main motivation of the paper and also acceptable as this is a short paper for MIDL)

Overall, the work is technically sound and practically motivated and will generate important discussions during MIDL meeting.

**Summary:**

This work proposes an axis-wise ridge regression blending strategy to improve 2D–3D registration for cochlear implant surgery by combining predictions from five complementary deep learning models, which were proposed previously by the authors. The method partitions models based on their strengths across degrees of freedom and learns separate regression weights for translation and rotation components while preserving cross-axis information. Evaluations are performed  using leave-one-out cross-validation on a small patient dataset, the approach achieves lower translation errors compared to individual models and alternative blending strategies.

**Strengths:**

Addresses a clinically important problem in surgical navigation and augmented reality for cochlear implant surgery.

Clear motivation: different models perform well on different degrees of freedom, justifying a hybrid approach.

Well-suited for small datasets; does not require GPU training, increasing practical applicability.

Demonstrates consistent improvements over individual baselines for translation components.

**Weaknesses:**

Limited methodological novelty: the approach is essentially a structured ensembling strategy using standard ridge regression, with only a modest contribution in axis-wise partitioning.

Very small dataset (7 patients in LOOCV) significantly limits statistical power and raises concerns about overfitting and generalizability.

Evaluation is restricted to variations of the same baseline methods; no comparison with broader state-of-the-art 2D–3D registration techniques (e.g., optimization-based, differentiable rendering, or transformer-based approaches).

Improvements are inconsistent across all degrees of freedom, with limited or no gains in rotation accuracy, which is critical for surgical guidance.

No statistical significance testing is provided to support performance improvements.

Lack of runtime and latency analysis, which is important for real-time surgical applications.

**Justification Of Rating:**

see above

---

### Decision · Program_Chairs · 2026-05-08

Accept (Poster)